# Effective Rank Analysis and Regularization for Enhanced 3D Gaussian Splatting

**Junha Hyung**[1]     **Susung Hong**[4]     **Sungwon Hwang**[1]     **Jaeseong Lee**[1]
**Jaegul Choo**[1†]     **Jin-Hwa Kim**[2,3†]

[1]KAIST     [2]NAVER AI Lab     [3]SNU AIIS     [4]Korea University

## Abstract

3D reconstruction from multi-view images is one of the fundamental challenges in computer vision and graphics. Recently, 3D Gaussian Splatting (3DGS) has emerged as a promising technique capable of real-time rendering with high-quality 3D reconstruction. This method utilizes 3D Gaussian representation and tile-based splatting techniques, bypassing the expensive neural field querying. Despite its potential, 3DGS encounters challenges such as needle-like artifacts, suboptimal geometries, and inaccurate normals caused by the Gaussians converging into anisotropic shapes with one dominant variance. We propose using the effective rank analysis to examine the shape statistics of 3D Gaussian primitives, and identify the Gaussians indeed converge into needle-like shapes with the effective rank 1. To address this, we introduce the effective rank as a regularization, which constrains the structure of the Gaussians. Our new regularization method enhances normal and geometry reconstruction while reducing needle-like artifacts. The approach can be integrated as an add-on module to other 3DGS variants, improving their quality without compromising visual fidelity. The project page is available at https://junhahyung.github.io/erankgs.github.io/.

## 1 Introduction

Creating 3D models from multiple images is a central challenge in computer vision and graphics. Neural Radiance Fields (NeRF) [20] have revolutionized this area by demonstrating remarkable capabilities in novel view synthesis through implicit neural fields and differentiable rendering techniques. Despite their impressive 3D reconstruction quality, the training and rendering processes of NeRF-based methods are computationally intensive, posing significant challenges for real-time applications. To improve training and rendering efficiency, various acceleration techniques, such as baking with shell [12, 32] and grid representations [5, 21], have been introduced. While these solutions enhance efficiency to some extent, there are still limitations for real-time interactive scenarios.

Recently, 3D Gaussian Splatting (3DGS) has emerged as a promising technique capable of real-time rendering with high-quality results. This method utilizes 3D Gaussian representations and tile-based splatting techniques instead of expensive neural field querying, making it feasible to apply the technique in practical applications. This opens up new possibilities in areas that require faster rendering, such as virtual and augmented reality, gaming, and real-time avatars.

However, despite its potential, 3DGS encounters several challenges in terms of geometry reconstruction, including noisy rendering results with needle-like artifacts, especially in novel and extreme views far from the training images. These issues stem from the primitive-based nature of 3DGS, where individual primitives lack geometric constraints.

---

[†]Corresponding authors

38th Conference on Neural Information Processing Systems (NeurIPS 2024).

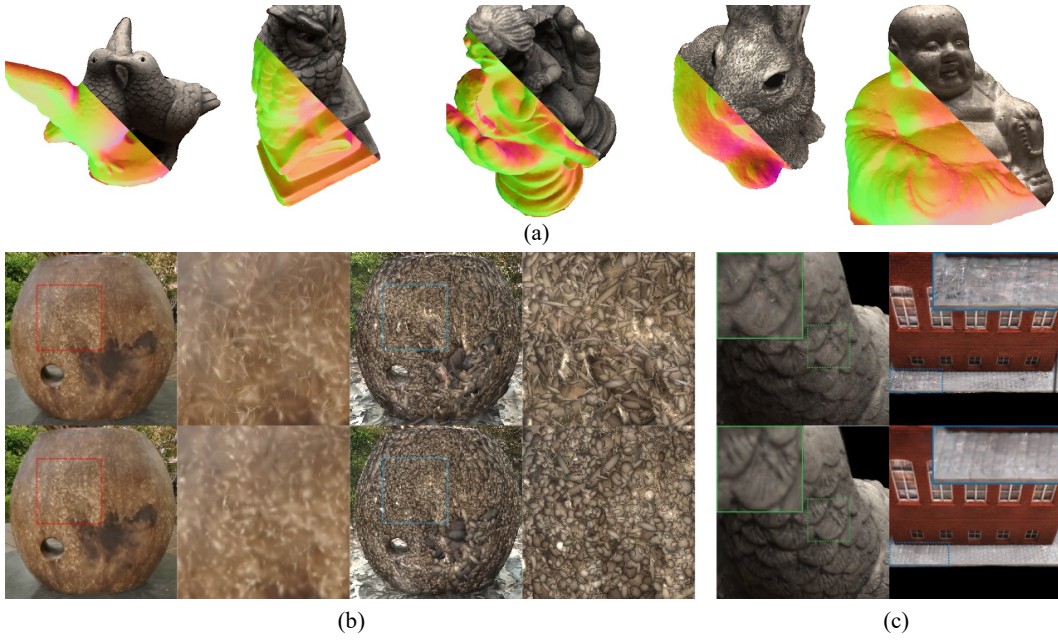

(a)

(b)                                                          (c)

Figure 1: (a) Qualitative results on novel view synthesis and normal reconstruction on the DTU [14] dataset. (b) and (c) show novel view synthesis comparisons on the Mip-NeRF360 [2] and DTU datasets, respectively. The top row shows novel view renderings of 3DGS, and the bottom row shows renderings of 3DGS with effective rank regularization. While naive 3DGS presents needle-like artifacts, our regularization term mitigates these artifacts in novel views.

For accurate geometry reconstruction, it is well known that the density field should be concentrated near the surface [30]. To this end, previous efforts, such as SuGaR [10], have focused on regularizing the 3D Gaussians to be flatter, *i.e.*, regularizing the primitives into anisotropic Gaussians with one of its variance very small. Similarly, 2DGS [13] utilizes 2D Gaussians instead of 3D Gaussians to force this effect.

While the flatness of Gaussians is necessary for proper alignment with the surface, we argue that flatness alone is insufficient for accurately representing surface geometry. Gaussians should also avoid being needle-like or highly anisotropic, where one variance dominates the others. Needle-like Gaussians hinder accurate reconstruction as they cover only a negligible portion of the surface and produce spiky artifacts. Instead, disk-like Gaussians are preferable as they cover meaningful areas and contribute effectively to surface reconstruction.

However, existing methods fail to adequately differentiate between disk-like and needle-like Gaussians, as both can exhibit one scale that is near or exactly zero. Empirically, we find that most Gaussians converge into anisotropic forms, becoming needle-like with small scales along two axes due to the absence of appropriate regularization mechanisms.

To directly examine the shape statistics (whether their geometries are disk-like or needle-like) of 3D Gaussian primitives and understand their structural changes during training, we first propose performing the effective rank analysis on the covariance matrices of Gaussians. The effective rank [25], which is a real-valued and differentiable extension of the integer rank, can be used to monitor the training dynamics and structural transformations of Gaussian primitives. Indeed, our analysis reveals that the effective ranks of Gaussians approach an effective rank of 1 (*erank*-1), resulting in needle-like shapes in 3DGS and other methods, such as SuGaR [10] and 2DGS [13].

Additionally, we propose using the effective rank as a regularization term to constrain the structure of the Gaussians. The differentiable nature of effective rank, with its concave logarithmic term providing stable gradients, makes it directly applicable to continuous optimization problems. Our new regularization method enhances normal and geometry reconstruction while reducing needle-like

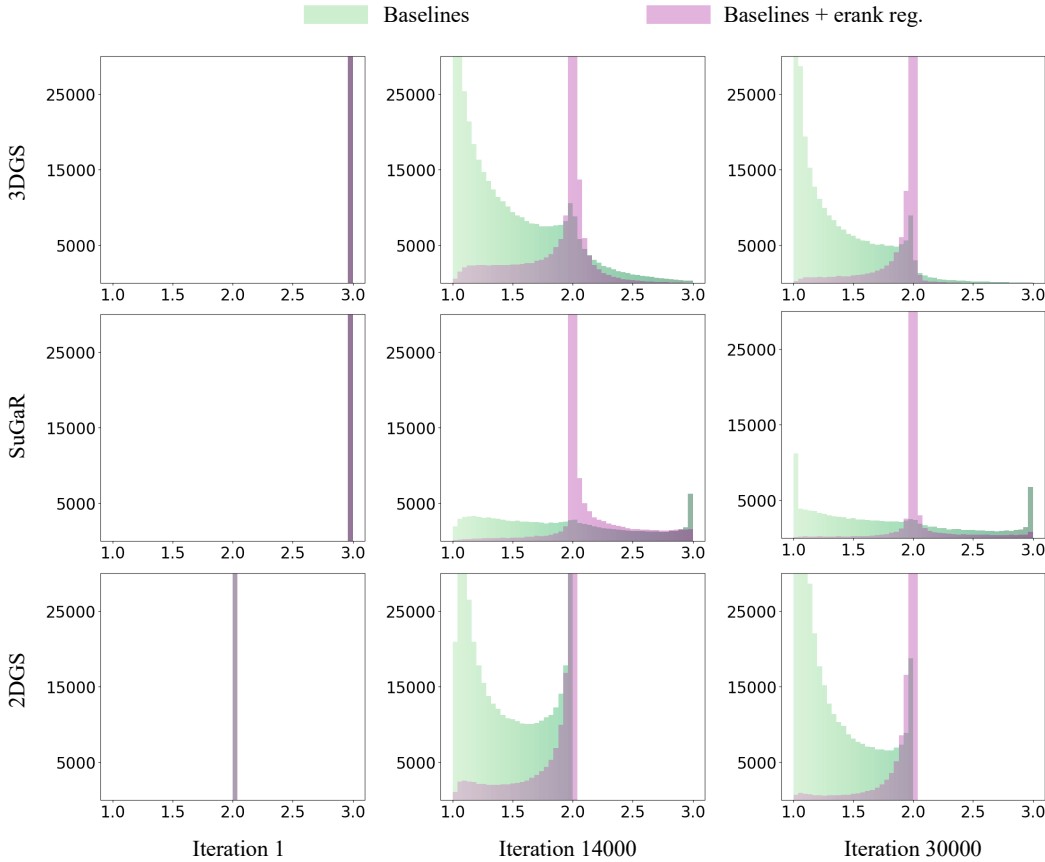

Figure 2: (Green): The effective rank histograms for baseline methods 3DGS [16], SuGaR [10], and 2DGS [13], showing that Gaussian ranks are not optimally constrained for geometry reconstruction. (Purple): The regularization term properly constrains the Gaussians, flattening them while preventing convergence into needle-like shapes.

artifacts, particularly in novel view scenarios. Furthermore, our effective rank regularization can be applied as an add-on module to other 3DGS variants, improving their quality.

The main contributions of our work are as follows:

- We are firstly analyzing the dynamics of Gaussian primitive structures using the effective rank in the optimizing process, discovering that Gaussians converge into anisotropic forms with one dominant variance.
- We propose an effective rank regularization method that alleviates needle-like artifacts in 3DGS rendering and improves geometric reconstruction.
- Our approach is an add-on module that can be integrated with other 3DGS variants, and demonstrate that our method enhances 3D geometry reconstruction without compromising visual quality.

## 2   Related work

**Novel view synthesis**   NeRFs [20] have revolutionized photo-realistic rendering from novel viewpoints by introducing a neural implicit representation of 3D scenes. This approach uses high-frequency positional encoding and differentiable volume rendering to achieve unprecedented realism. Enhancements to NeRF address challenges like anti-aliasing [1, 3], parameterizing unbounded scenes [2, 37], and training from in-the-wild images [19, 8, 29] through probabilistic transience modeling. Further

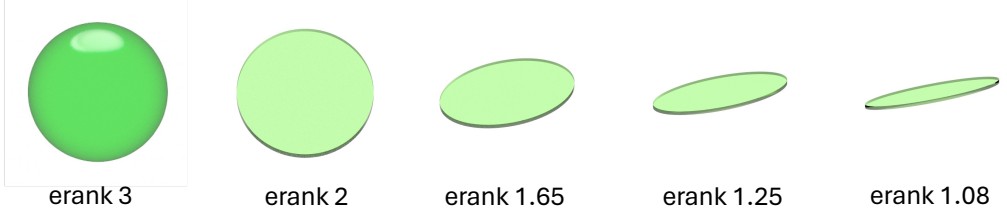

Figure 3: Real-scale visualization of a 3D sphere and 2D disks and their effective ranks.

improvements reduce training time and enhance rendering quality by incorporating low-rank tensor components [5].

Other research efforts have aimed for real-time rendering using alternative implicit models that do not rely on MLPs. Notable examples include sparse voxel grids [9] and multi-resolution hash encoding [21]. Despite these advancements, ray tracing methods are inherently slower than rasterization. To address this, 3DGS [16] introduced a point-based rasterization technique for real-time, high-fidelity view synthesis. Inspired by EWA Volume Splatting [39], 3DGS uses a fully differentiable pipeline, representing 3D scenes with 3D Gaussians and performing volume splatting to known camera poses for rasterization.

**Surface reconstruction**     Surface reconstruction is a critical area in computer vision and graphics, aiming to recreate 3D shapes and structures from 2D images or other data forms. Among recent innovations, NeuS [30] leverages volume rendering and signed distance functions (SDF) for high-fidelity reconstructions. NeuS2 [31] significantly improves training speed and extends modeling capacity to dynamic scenes. UNISURF [22] integrates implicit surface models and radiance fields for both surface and volume rendering. VolSDF [33] models volume density as a function of geometry, achieving high-quality geometry reconstructions. Neuralangelo [18] uses multi-resolution hash grids and neural surface rendering to recover detailed structures. BakedSDF [34] introduces a hybrid neural volume-surface representation optimized for mesh extraction.

**3D Gaussian Splatting**     Recent advancements in 3DGS have further propelled surface reconstruction. NeuSG [6] refines surface details using 3DGS and neural implicit models. SuGaR [10] focuses on mesh extraction with SDF-based regularization and Poisson reconstruction. 2DGS [13] collapses 3D volumes into 2D Gaussian disks for view-consistent geometry and detailed mesh reconstruction. GaussianShader [15] enhances rendering quality in reflective surfaces using a shading function on 3D Gaussians. GOF [36] utilizes ray-Gaussian intersection for density estimation and geometric regularization. GIR [28] employs 3D Gaussians for inverse rendering, enabling accurate estimation of material properties, illumination, and geometry. These advancements showcase the potential of 3DGS for high-speed, detailed, and versatile surface reconstructions.

## 3   Preliminaries

### 3.1   3D Gaussian splatting

3DGS [16] represents a scene with a set of learnable 3D Gaussian primitives $\{\mathcal{G}_k \mid k = 1, \cdots, K\}$, where each 3D Gaussian $\mathcal{G}_k$ consists of mean $\boldsymbol{\mu}_k \in \mathbb{R}^{3 \times 1}$, covariance $\boldsymbol{\Sigma}_k \in \mathbb{R}^{3 \times 3}$, point opacity $\alpha_k \in [0, 1]$ and view-dependent color $c_k$ in spherical harmonics. Covariance matrix $\boldsymbol{\Sigma}_k = \mathbf{R}_k \mathbf{S}_k \mathbf{S}_k^\top \mathbf{R}_k^\top$ is positive semi-definite, where $\mathbf{S}_k = \mathrm{diag}(\mathbf{s}_k)$ is a scaling matrix, $\mathbf{s}_k = (s_{k1}; s_{k2}; s_{k3}) \in \mathbb{R}^{3 \times 1}$ is a scale parameter, and $\mathbf{R}_k \in \mathbb{R}^{3 \times 3}$ is a rotation matrix parameterized by a quaternion. A 3D Gaussian primitive can be represented in 3D space as:

$$\mathcal{G}_k(\mathbf{x}) = e^{-\frac{1}{2}(\mathbf{x} - \mu_k)^T \boldsymbol{\Sigma}_k^{-1}(\mathbf{x} - \mu_k)}. \tag{1}$$

The primitives are then rasterized via differentiable volume splatting. Specifically, a 3D Gaussian is projected to 2D screen space as $\boldsymbol{\Sigma}_k^{'} = \mathbf{J}\mathbf{W}\boldsymbol{\Sigma}_k \mathbf{W}^\top \mathbf{J}^\top$, where $\mathbf{W}$ is a world-to-camera transform and $\mathbf{J}$ is the Jacobian of the affine approximation of the projection matrix [39]. The covariance and mean

of the projected Gaussian $\mathcal{G}_k^{2D}(\mathbf{x})$ are then obtained by removing the third column and row of $\boldsymbol{\Sigma}_k^{'}$ and simply projecting $\mu_k$ to screen space, respectively. Finally, the Gaussians are alpha-blended in the order of depth as:

$$\mathbf{c}(\mathbf{u}) = \sum_{k=1}^{K} c_k \alpha_k \prod_{j=1}^{k-1} (1 - \alpha_j \mathcal{G}_j^{2D}(\mathbf{u})), \tag{2}$$

where $\mathbf{u}$ is a screen space coordinate. The rendered images are supervised with photometric loss $L$ for 3D primitive optimization similar to NeRF [20].

As Gaussians are initialized by sparse SfM points, Adaptive Density Control (ADC) is designed for densification during optimization. Specifically, ADC subsamples and splits Gaussians that satisfy the condition:

$$\left\| \frac{\partial L}{\partial \mathbf{u}} \right\|_2 = \left\| \sum_{i \in \mathcal{P}} \frac{\partial L}{\partial \mathbf{p}_i} \frac{\partial \mathbf{p}_i}{\partial \mathbf{u}} \right\|_2 > \tau, \tag{3}$$

where $\mathcal{P}$ and $\mathbf{p}_i$ denote a set of pixel indices and the $i$-th pixel, respectively, and $\tau$ is a predefined threshold. The intuition behind Eq. 3 is that regions not yet well reconstructed exhibit large view-space positional gradients. This occurs because the optimization process attempts to move the Gaussians to correct these areas, so densifying such Gaussians can effectively increase expressibility.

### 3.2 Effective rank

Consider a real-valued non-all-zero $M \times N$ matrix $\mathbf{A}$. The singular value decomposition (SVD) of $\mathbf{A}$ can be expressed as $\mathbf{A} = \mathbf{U}\mathbf{D}\mathbf{V}$, where $\mathbf{U}$ and $\mathbf{V}$ are unitary matrices of sizes $M \times M$ and $N \times N$ respectively, and $\mathbf{D}$ is a diagonal matrix of size $M \times N$ containing the real positive singular values in descending order:

$$\sigma_1 \geq \sigma_2 \geq \cdots \sigma_L \geq 0, \tag{4}$$

where $L = \min\{M, N\}$. The *singular value distribution* is then defined as

$$q_i = \frac{\sigma_i}{\|\boldsymbol{\sigma}\|_1}, \text{for } i = 1, 2, \cdots, L, \tag{5}$$

where $\boldsymbol{\sigma} = (\sigma_1, \sigma_2, \cdots, \sigma_L)^T$, and $\| \cdot \|_1$ denotes $\ell_1$-norm.

**Definition 1** (Effective rank). The effective rank [25] of the matrix $\mathbf{A}$ is concisely defined as $\text{erank}(\mathbf{A}) = \exp\{H(q_1, q_2, \cdots, q_L)\}$, where $H(q_1, q_2, \cdots, q_L)$ is the Shannon entropy given by

$$H(q_1, q_2, \cdots, q_L) = -\sum_{i=1}^{L} q_i \log q_i. \tag{6}$$

## 4 Method

In Section 4.1, we introduce the effective rank analysis to inspect the geometries of Gaussians of 3DGS and its variants, shedding light on their underlying structures. Based on the findings from our effective rank analysis, we propose a novel effective rank regularization method in Section 4.2.

### 4.1 Effective rank analysis of 3D Gaussians

We propose to analyze the effective rank to investigate the structural dynamics of individual 3D Gaussians by calculating the effective rank of the covariance matrix of the Gaussians. The covariance matrix of the 3D Guassians is defined as $\boldsymbol{\Sigma}_k = \mathbf{R}_k \mathbf{S}_k \mathbf{S}_k^T \mathbf{R}_k^T$, and the diagonal matrix after SVD is $\mathbf{D} = \mathbf{S}_k \mathbf{S}_k^T$, with real positive singular values in a descending order as follows:

$$s_1^2 \geq s_2^2 \geq s_3^2 > 0, \tag{7}$$

where we omit subscript $k$ of $\mathbf{s}_k$ for brevity.

Accordingly, we can derive the effective rank of a 3D Gaussian $\mathcal{G}_k$ with the covariance matrix $\boldsymbol{\Sigma}_k$. The entropy term is $H(\mathcal{G}_k) := H(q_1, q_2, q_3) := -\sum_{i=1}^{3} q_i \log q_i$, with

$$\mathbf{q} = (q_1, q_2, q_3) = \left( \frac{s_1^2}{S}, \frac{s_2^2}{S}, \frac{s_3^2}{S} \right), \quad \text{where} \quad S = \sum_{i=1}^{3} s_i^2, \tag{8}$$

and the effective rank of a 3D Gaussian $\mathcal{G}_k$ with the covariance matrix $\boldsymbol{\Sigma}_k$ is defined as follows:

$$\text{erank}(\mathcal{G}_k) := \exp\{H(\mathcal{G}_k)\}. \tag{9}$$

The effective rank, being a differentiable extension of an integer rank, is a suitable tool for geometric analysis of 3D Gaussians since it jointly considers all of the scale parameters and can identify the relative scales of the three axes. The advantage of effective rank becomes more apparent when compared to recent works that only analyze individual or pair-wise variances of the 3D Gaussians [15]. Such approaches do not fully represent the geometry of Gaussians, potentially leading to planar and needle-like Gaussians being categorized together. For better understanding, we visualize the effective ranks of a sphere and 2D disks in Fig. 3.

With the distinct advantage of our approach, we can differentiate between needle-like Gaussians, which have effective ranks close to 1, and planar disk-like Gaussians. To reconstruct a scene with an accurate surface, we need Gaussians that represent a plane that aligns and concentrates well with the surface [30]. Ideally, 3D Gaussians with $\text{erank}(\mathcal{G}_k) \approx 2$ are preferred, but Gaussians with an effective rank smaller than 2 are also required for representing thin and elongated objects and patterns. However, the needle-like Gaussians with $\text{erank}(\mathcal{G}_k) \approx 1$ are undesirable because they account for a negligible region of the surface and produce degenerate results in novel views.

The first row of Fig. 2 (green graph) shows the effective rank histogram for 3DGS during training. As the model converges, the number of 3D Gaussians with $\text{erank}(\mathcal{G}_k) \approx 1$ increases, indicating overfitting without improvements in PSNR and Chamfer distance metrics (metrics are provided in the Appendix A.5, Table 5). This indicates that the majority of "flat" Gaussians (singular values close to 0) are actually needle-like ($\text{erank}(\mathcal{G}_k) \approx 1$), rather than disk-like ($\text{erank}(\mathcal{G}_k) \approx 2$). It is also interesting to note that 3DGS naturally forms a small mode at $\text{erank}(\mathcal{G}_k) = 2$, indicating an observed preference that can be further strengthened with our regularization.

Despite having different geometric constraints on the Gaussians, SuGaR [10] (the second row in Fig. 2) and 2DGS [13] (the third row in Fig. 2) also exhibit a similar tendency to have a large amount of needle-like Gaussians with a single dominant variance along an axis. Notice that all Gaussians in 2DGS start with an effective rank of exactly 2, but the majority still fail to remain disk-shaped and instead become needle-like 2D Gaussians.

## 4.2 Optimization

The real-valued and differentiable nature of the effective rank allows us to utilize it as a regularization objective to impose structural constraints on 3D Gaussians. Specifically, our goal is to keep the effective rank of 3D Gaussians below 2, thereby promoting planar shapes, while penalizing Gaussians with an effective rank close to 1 to minimize needle-like artifacts. Although disk-like Gaussians with $\text{erank}(\mathcal{G}_k) \approx 2$ are preferred, shapes with $\text{erank}(\mathcal{G}_k) < 2$ are also essential for representing complex geometries. We propose an effective rank regularization term that increases exponentially as the effective rank nears 1, strongly penalizing such Gaussians:

$$\mathcal{L}_{\text{erank}} = \sum_k \lambda_{\text{erank}} \max(-\log(\text{erank}(\mathcal{G}_k) - 1 + \epsilon), 0) + s_3, \tag{10}$$

where $\epsilon = 1 \times 10^{-5}$ ensures numerical stability, and $s_3$ is the smallest scale parameter of $\mathcal{G}_k$. The regularization effectively constrains the effective rank of Gaussian primitives when added to the baselines, as shown in the purple graphs of Fig. 2. Also, the regularization is scheduled to be applied from 7000-iteration, adhering to the coarse-to-fine training paradigm, which enables stable training upon early iterations with $\text{erank}(\mathcal{G}_k) > 2$ Gaussians.

**ADC algorithm** We adopt the revised version of the densification algorithm presented in [4, 36], which densifies Gaussians based on the summation of norms instead of the norm of the summation in

Table 1: Chamfer distance and PSNR report on DTU dataset. +e denotes the erank regularization.

| Method | 24 | 37 | 40 | 55 | 63 | 65 | 69 | 83 | 97 | 105 | 106 | 110 | 114 | 118 | 122 | Mean | Std. | PSNR |
|---|---|---|---|---|---|---|---|---|---|---|---|---|---|---|---|---|---|---|
| 3DGS | 2.14 | 1.53 | 2.08 | 1.68 | 3.49 | 2.21 | 1.43 | 2.07 | 2.22 | 1.75 | 1.79 | 2.55 | 1.53 | 1.52 | 1.50 | 1.96 | 0.52 | 32.82 |
| 3DGS+e | **0.86** | **0.77** | **0.88** | **0.52** | **1.29** | **1.44** | **0.96** | **1.30** | **2.09** | **0.72** | **0.87** | **1.40** | **0.88** | **0.94** | **0.66** | **1.04** | 0.39 | **33.09** |
| SuGaR | 1.47 | 1.33 | 1.13 | 0.61 | 2.25 | 1.71 | 1.15 | 1.63 | 1.62 | 1.07 | **0.79** | 2.45 | 0.98 | **0.88** | 0.79 | 1.33 | 0.52 | 31.59 |
| SuGaR+e | **0.86** | **0.78** | **0.89** | **0.53** | **1.28** | **1.45** | **0.87** | **1.31** | **1.60** | **0.72** | 0.86 | **1.45** | **0.87** | 0.94 | **0.66** | **1.00** | 0.33 | **31.76** |
| 2DGS | 0.48 | 0.91 | **0.39** | 0.39 | 1.01 | **0.83** | 0.81 | 1.36 | 1.27 | 0.76 | 0.70 | 1.40 | 0.40 | 0.76 | 0.52 | 0.80 | 0.33 | 32.43 |
| 2DGS+e | **0.46** | **0.86** | 0.39 | 0.40 | **0.96** | 0.84 | 0.81 | **1.29** | **1.19** | 0.72 | 0.70 | **1.32** | 0.40 | **0.75** | **0.50** | **0.77** | 0.30 | **32.57** |
| GOF | 0.50 | 0.82 | 0.37 | **0.37** | 1.12 | 0.78 | 0.73 | 1.18 | 1.29 | 0.71 | 0.77 | 0.90 | 0.44 | 0.69 | 0.49 | 0.74 | 0.28 | 32.88 |
| GOF+e | **0.45** | **0.66** | **0.32** | 0.42 | **0.97** | 0.78 | **0.64** | **1.13** | **1.22** | **0.64** | **0.62** | **0.70** | **0.40** | **0.53** | **0.48** | **0.66** | 0.26 | **33.01** |

Table 2: Ablation study result of our method on DTU dataset. (a): the fixed densification (ADC) algorithm, (b): erank regularization, (c): optional bag of tricks discussed in the Appendix.

| Method | 24 | 37 | 40 | 55 | 63 | 65 | 69 | 83 | 97 | 105 | 106 | 110 | 114 | 118 | 122 | Mean | PSNR |
|---|---|---|---|---|---|---|---|---|---|---|---|---|---|---|---|---|---|
| 3DGS | 2.14 | 1.53 | 2.08 | 1.68 | 3.49 | 2.21 | 1.43 | 2.07 | 2.22 | 1.75 | 1.79 | 2.55 | 1.53 | 1.52 | 1.50 | 1.96 | 32.82 |
| +a | 1.24 | 0.97 | 1.09 | 0.62 | 1.45 | 1.55 | 1.14 | 1.58 | 2.31 | 0.92 | 1.08 | 1.72 | 1.02 | 1.22 | 0.97 | 1.26 | 32.97 |
| +a+b | 0.85 | 0.77 | 0.88 | 0.51 | 1.21 | 1.45 | 0.96 | 1.30 | 2.09 | 0.72 | 0.86 | 1.45 | 0.87 | 0.94 | 0.66 | 1.03 | **33.09** |
| +a+b+c | **0.45** | **0.66** | **0.32** | **0.42** | **0.97** | **0.78** | **0.64** | **1.13** | **1.22** | **0.64** | **0.62** | **0.70** | **0.40** | **0.53** | **0.48** | **0.66** | 33.01 |

Eq. 3 (further details in Appendix A.4). This change is particularly important for our regularization method. Disk-like Gaussians, unlike needle-like ones, often fail to satisfy the splitting criterion set by Eq. 3 because their axes with larger variances produce smaller gradient signals per pixel. Moreover, since disk-like Gaussians typically cover more pixel space, unaligned signals tend to cancel each other out. In contrast, the revised densification algorithm facilitates the splitting of disk-like Gaussians. Notably, due to their superior ability to reconstruct surfaces compared to needle-like Gaussians, our method requires approximately 10% fewer Gaussians than the baseline [16].

## 5 Experiments

We evaluate the effective rank regularization, comparing its performance as an add-on to baseline models. Additionally, we analyze the contributions of different components of the method.

### 5.1 Implementation

The regularization hyperparameter $\lambda_{\mathrm{erank}} = 0.01$ is used for all training. For other components belonging to the baselines, we use the same settings as described in the corresponding papers. All experiments are conducted on a Tesla V100 GPU. For mesh extraction, the truncated signed distance function (TSDF) fusion with Open3D [38] is used, with details in the Appendix A.3.

### 5.2 Comparison

**Dataset**  We evaluate our model on the DTU [14] and Mip-NeRF360 [2] datasets. The DTU dataset consists of 15 forward-facing bounded scenes with a resolution of $1600 \times 1200$. Following prior standards [13, 36], we downsample the images to a resolution of $800 \times 600$. The DTU dataset is used for evaluating both geometry reconstruction (using Chamfer distance) and novel view synthesis. The Mip-NeRF360 dataset comprises 9 indoor and outdoor scenes with images at a resolution of $1600 \times 1050$ and is used exclusively for novel view synthesis evaluation. For novel view synthesis, the images are split into training and test sets, while the entire set of images is used for geometry reconstruction. COLMAP [26, 27] is used to initialize point clouds for the baselines.

**Baselines**  Our method is applicable to other baselines as an add-on term. Therefore, we compare baselines with and without our regularization. We choose SuGaR, 2DGS, and GOF as our baselines, works that focus on better geometry reconstruction, along with the original 3DGS. All of the experiments are performed with the proposed setting of the original paper.

**Geometry reconstruction**  Table 1 presents the quantitative results of geometry reconstruction on the DTU dataset. We report the Chamfer distance for each scene, along with the mean Chamfer

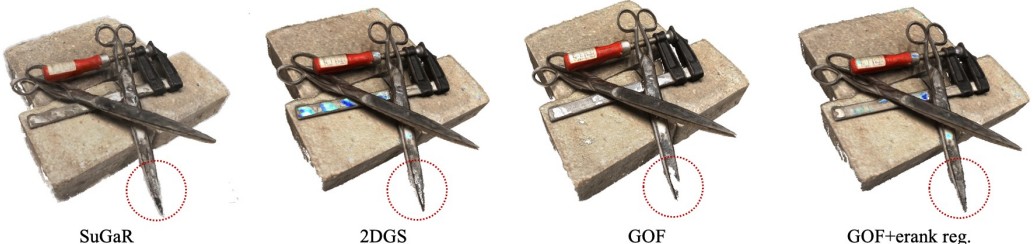

SuGaR      2DGS      GOF      GOF+erank reg.

Figure 4: Visualization of the reconstructed mesh using TSDF. Baseline methods often exhibit empty holes, while our regularization term enforces disk-like Gaussians, reducing such artifacts and improving surface reconstruction.

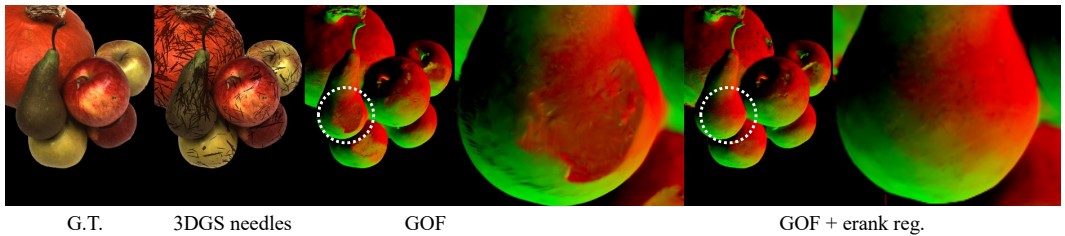

G.T.      3DGS needles      GOF      GOF + erank reg.

Figure 5: Normal reconstruction results on the DTU dataset. Needle-like Gaussians often leave empty holes or transparent regions, resulting in hollow or incomplete reconstructions, as seen on the pear surface. The effective rank regularization significantly mitigates these artifacts, leading to more accurate geometry reconstruction.

distance and mean PSNR. The "+e" symbol indicates the addition of the effective rank regularization (with fixed densification) to the baseline methods.

The results show that methods enhanced with our add-on term outperform the baselines. Notably, applying our regularization to 3DGS (3DGS+e) results in a significant improvement in geometry reconstruction, demonstrating the effectiveness of the regularization. This supports our hypothesis that reducing needle-like Gaussians and achieving flatness as in Fig. 2 improves performance. Additionally, the figure shows that SuGaR contains both needle-like and non-planar Gaussians with effective ranks greater than 2. By attaining flatness and removing spikes through effective rank regularization, we achieve a substantial performance gain for SuGaR (SuGaR+e).

GOF and 2DGS already incorporate well-designed regularization terms, such as depth distortion loss [13, 2], to align Gaussians with surfaces and enhance geometry reconstruction. Furthermore, 2DGS explicitly uses 2D Gaussians as their primitive, inherently achieving planarity. Nonetheless, our method prevents Gaussians from converging into needles in both approaches (and enforces flatness in GOF), resulting in performance gains.

Fig. 4 shows mesh reconstruction results, where baseline methods often exhibit empty holes in the reconstructed meshes. Our regularization term enforces disk-like Gaussians, reducing such holes and proving advantageous for surface reconstruction.

Fig. 5 and the first row of Fig. 1 display normal reconstruction results. In Fig. 5, the resulting image from GOF shows spiky artifacts and a hollow surface on the pear. Similarly to the mesh results, needle-like Gaussians often fail to cover the entire area, leaving empty holes or transparent regions, resulting in hollow or incomplete reconstructions. The effective rank regularization mitigates these noisy artifacts, leading to a more accurate reconstruction of the underlying geometry.

**Novel view synthesis**     Since 3D reconstruction from 2D images is an ill-posed problem, Gaussians tend to overfit to the training views, converging into needle-like shapes and causing spiky artifacts in test views, as shown in Fig. 1 (b), (c), and Fig.6. For better understanding, we visualize Gaussians with $\text{erank}(\mathcal{G}_k) < 1.02$ (scale ratio of approximately 20:1 or larger) in red. Our method mitigates overfitting and the resulting artifacts by enforcing structural priors on the Gaussians.

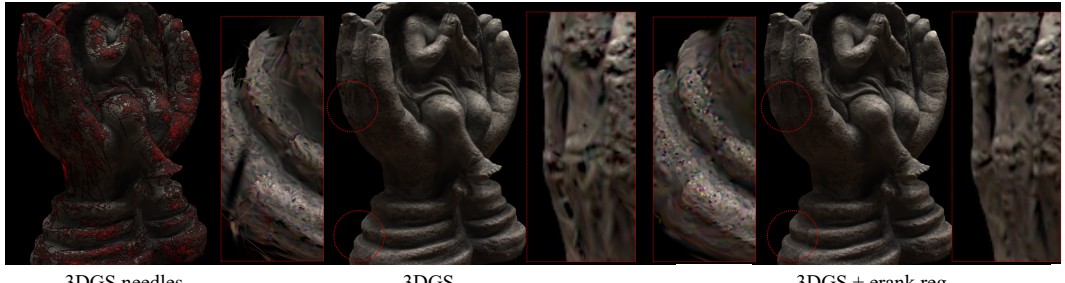

| 3DGS needles | 3DGS | 3DGS + erank reg. |

Figure 6: Qualitative comparison on DTU dataset. Gaussians with erank$(\mathcal{G}_k) < 1.02$ are visualized in red. Our regularization term mitigates needle-like artifacts in novel views.

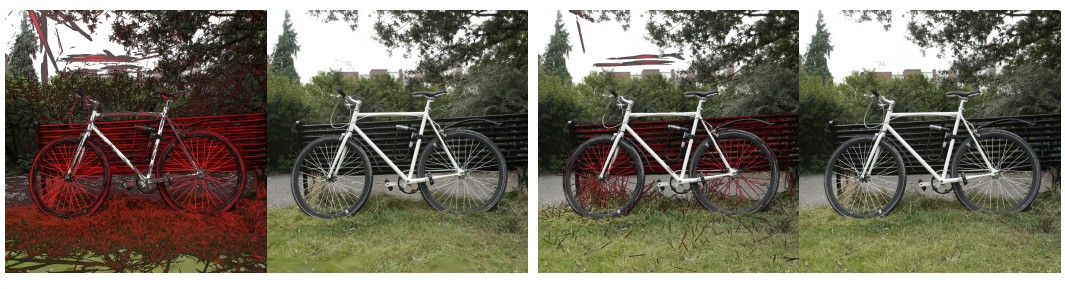

| 3DGS ( PSNR: 25.1 / 1.4 GB) | 3DGS + erank reg. (PSNR: 25.5 / 1.2 GB) |

Figure 7: Qualitative comparison on Mip-NeRF360 dataset. Our method effectively represents thin objects, achieving better visual quality and compactness

Furthermore, as seen in Fig. 7, our method adaptively preserves some elongated Gaussians when necessary, allowing the representation of slender structures. The results indicate that while 3DGS heavily relies on needle-like Gaussians to represent the scene, our method limits their use to only when required, leading to improved novel view synthesis performance.

We also provide quantitative results in Table 1, where we report the average PSNR for the DTU dataset. Results for Mip-NeRF360 are reported in Table 3 in the Appendix A.5. While many geometry regularization techniques degrade visual quality, our method does not exhibit this trade-off and actually shows slight improvements by properly constraining the shape of the Gaussians.

**Efficiency**    As shown in Fig. 7 and Table 4, the efficacy of disk-like Gaussians in 3D reconstruction, compared to needle-like Gaussians, leads to a better memory footprint. The average storage usage for the DTU and Mip-NeRF360 datasets is reported in Table 4 (Appendix A.5).

### 5.3   Ablations

Our method comprises two key components: (a) the fixed densification (ADC) algorithm and (b) the effective rank regularization. We performed an ablation study on these components to observe their performance gains compared to the naive 3DGS method. Table 2 shows the Chamfer distance and PSNR measured on the DTU dataset. The results indicate that both components contribute to performance gains in geometry reconstruction and novel view synthesis tasks. Additionally, incorporating techniques such as depth distortion loss [13, 36] can further enhance the best-performing model (row +a+b+c). These techniques are discussed in the Appendix A.2.

## 6   Conclusion

**Limitations**    Our regularization term constrains individual Gaussians but does not account for the local and global structure of the scene. Thus, it may be beneficial to pair our method with structure-aware regularizations, such as the depth distortion loss [13], which considers the Gaussians

along the ray collectively. Another limitation is the manual selection of the hyperparameter $\lambda_{\text{erank}}$. While our chosen hyperparameter works well for the scenes used in our evaluation, it may not be optimal for extreme scenes dominated by thin objects and structures.

## Acknowledgments and Disclosure of Funding

Junha Hyung and Susung Hong conducted this work during the internship at NAVER AI Lab. The NAVER Smart Machine Learning (NSML) platform [17] had been used for experiments. This work was supported by KAIST-NAVER hypercreative AI center. This work was partly supported by Institute for Information & communications Technology Promotion (IITP) grant funded by the Korea government (MSIT) (No.RS-2019-II190075 Artificial Intelligence Graduate School Program (KAIST)). This work was also partly supported by the National Research Foundation of Korea (NRF) grant funded by the Korea government (MSIT) (No. NRF-2022R1A2B5B02001913).

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

# A    Appendix / supplemental material

## A.1    Broader impact

The broader impact of our work on 3D reconstruction lies in its potential to advance various fields such as virtual and augmented reality, medical imaging, and digital content creation by enabling more efficient and high-quality 3D model generation. However, as with any advanced technology, it also presents potential risks and avenues for misuse. For instance, enhanced 3D reconstruction techniques could be exploited to create deepfakes or unauthorized reproductions of proprietary designs, posing ethical and legal challenges. To mitigate these risks, we propose implementing strict usage guidelines to ensure the integrity and rightful use of 3D models. We aim to maximize the positive impact of our research while minimizing potential negative consequences.

## A.2    Additional regularization

For rendering normals, we add other regularization terms, such as depth distortion loss [13] and normal regularization, as proposed in [13, 36]. (We do not utilize these regularization terms for evaluating effective rank regularization as an add-on module in Tab. 1.) The depth distortion loss, which concentrates splats on a surface and mitigates floater artifacts, is given as

$$\mathcal{L}_d = \lambda_d \sum_{i,j} \omega_i \omega_j |z_i - z_j|, \tag{11}$$

where $\omega_i = \alpha_i \, \mathcal{G}_i(\mathbf{x}) \prod_{k=1}^{i-1} (1 - \alpha_k \, \mathcal{G}_k(\mathbf{x})))$ for the blending weight and $z_i$ is the depth of the intersection point of the $i-$th Gaussian, and $i, j$ are indexes over Gaussians contributing to a certain ray.

The normal regularization minimizes the difference between the rendered normal map $\bar{\mathbf{n}}$ of the splats and the gradient normals $\hat{\mathbf{n}}$ derived from the rendered depth map,

$$\mathcal{L}_\mathbf{n} = \lambda_n \left\| \bar{\mathbf{n}} - \hat{\mathbf{n}} \right\|, \tag{12}$$

which locally aligns the 3D Gaussians with the actual surfaces. Since the effective rank regularization does not account for the local and global structure of the scene, it is beneficial to pair our method with these structure-aware regularizations.

## A.3    Mesh extraction

We utilize the Truncated Signed Distance Function (TSDF) fusion for mesh extraction. The algorithm encodes the distance of any point in the voxel grid to the nearest surface, with the distance being truncated to a maximum value to limit the influence of faraway points. The sign of the distance function indicates whether the point is inside (negative) or outside (positive) the object. Multiple TSDFs are combined from different viewpoints to create a more accurate and complete 3D reconstruction, forming a coherent and comprehensive 3D model. The Marching Cubes algorithm is then used for triangulation.

## A.4    ADC fix

We adopt the revised version of the densification algorithm presented in [4, 36], which densifies Gaussians based on the summation of the norm instead of the norm of the summation in Eq. 3:

$$\sum_{i \in \mathcal{P}} \left\| \frac{\partial L}{\partial \mathbf{p}_i} \frac{\partial \mathbf{p}_i}{\partial \mathbf{u}} \right\|_2 > \tau. \tag{13}$$

As discussed in the main paper, this approach is crucial to our regularization because disk-like Gaussians typically cover more screen space and receive gradient signals from various pixels, which can cancel out when summed. The revised algorithm ensures the effective splitting of Gaussians with our regularization. However, due to the efficiency of disk-like Gaussians in surface reconstruction, our method requires about 10% fewer Gaussians compared to the baseline [16].

Table 3: Quantitative results on Mip-NeRF 360 [2] dataset.

| | Outdoor Scene | | | Indoor scene | | |
|---|---|---|---|---|---|---|
| | PSNR ↑ | SSIM ↑ | LPIPS ↓ | PSNR ↑ | SSIM ↑ | LIPPS ↓ |
| Mobile-NeRF [7] | 21.95 | 0.470 | 0.470 | - | - | - |
| BakedSDF [34] | 22.47 | 0.585 | 0.349 | 27.06 | 0.836 | 0.258 |
| BOG [23] | 23.94 | 0.680 | 0.263 | 27.71 | 0.873 | 0.227 |
| NeRF [20] | 21.46 | 0.458 | 0.515 | 26.84 | 0.790 | 0.370 |
| Deep Blending [11] | 21.54 | 0.524 | 0.364 | 26.40 | 0.844 | 0.261 |
| Instant NGP [21] | 22.90 | 0.566 | 0.371 | 29.15 | 0.880 | 0.216 |
| MERF [24] | 23.19 | 0.616 | 0.343 | 27.80 | 0.855 | 0.271 |
| MipNeRF360 [2] | 24.47 | 0.691 | 0.283 | 31.72 | 0.917 | 0.180 |
| 3DGS [16] | 24.64 | 0.731 | 0.234 | 31.13 | 0.920 | 0.189 |
| 3DGS+e (Ours) | 24.93 | 0.757 | 0.221 | 31.16 | 0.953 | 0.181 |

Table 4: Storage usage of our method, along with Chamfer distance, PSNR, and optimization time.

| Dataset | Method | CD ↓ | PSNR ↑ | Time ↓ | MB (Storage) ↓ |
|---|---|---|---|---|---|
| DTU | 3DGS | 1.96 | 32.82 | 11.2m | 113 |
| | 3DGS+e | 1.03 | 33.09 | 11.1m | 98 |
| Mip-NeRF360 | 3DGS | - | 27.52 | 41m | 734 |
| | 3DGS+e | - | 27.70 | 40m | 646 |

## A.5 Additional quantitative results

We report novel view synthesis results on the Mip-NeRF360 dataset in Table 3. The results show that our add-on regularization term improves the visual quality of 3DGS in terms of PSNR, SSIM, and LPIPS. Also, the method even shows comparable or slightly better performance compared to the NeRF variants with slow and computationally intensive rendering.

We report the training time of our method in Table 4. The training time for 3DGS on the DTU [14] dataset averages 11.2 minutes per scene. Adding the effective rank regularization with the densification fix incurs no overhead since the additional computation is compensated with a reduced number of Gaussians. Total training time is on average 11.1 minutes for the DTU dataset and 40 minutes for the Mip-NeRF360 dataset, on a single V100 GPU, reported in Table 4.

Also, with a reduced number of Gaussians, our method requires less memory and storage for scene representation, as in Table 4. While being more compact, our method outperforms baselines in terms of Chamfer distance and PSNR.

Table 5 demonstrates Chamfer distance and PSNR changes during the course of training, for the baselines shown in Fig. 2. Results are reported for the scene 37 of the DTU dataset. Needle-like Gaussians increase, but the performance plateaus, indicating overfitting. Additionally, different Gaussian structures with similar metrics suggest the heterogeneous nature of Gaussians in 3DGS and its variants. Also, the reported "Number of needles" corresponds to Gaussians with an effective rank smaller than 1.04. The results suggest that our regularization term effectively minimizes the number of needles without a visual quality trade-off.

We present per scene PSNR on the DTU dataset in Table 6. The mean PSNR is already shown in Table 1 and Table 2 of the main paper.

## A.6 Cause of needle-like Gaussians

While not directly related to our methodology, we investigate some reasons for the convergence of 3D Gaussians into anisotropic Gaussians with one dominant variance.

First, the scale of the 3D Gaussians is not properly constrained due to the dilation operation, which adds a small constant to screen space Gaussians [16] to ensure a minimum scale, as noted in Mip-

Table 5: Chamfer distance and PSNR changes during the course of training for the baselines shown in Fig. 2, for scene 37 of DTU dataset. Needle-like Gaussians increase, but the performance plateaus, indicating overfitting. Additionally, different Gaussian structures with similar metrics suggest the heterogeneous nature of Gaussians in 3DGS and its variants. Reported "Number of needles" correspond to Gaussians with effective rank smaller than 1.04.

| | CD↓ | | PSNR↑ | |
|---|---|---|---|---|
| Method | 15k | 30k | 15k | 30k |
| 3DGS | 1.5 | 1.53 | 27.00 | 26.98 |
| SuGaR | 1.21 | 1.23 | 23.64 | 23.52 |
| 2DGS | 0.89 | 0.88 | 24.89 | 24.87 |

| | Number of needles | | | PSNR↑ |
|---|---|---|---|---|
| | 0k | 15k | 30k | 30k |
| 3DGS | 0 | 3170 | 16320 | 26.93 |
| 3DGS+e | 0 | 28 | 23 | 27.21 |

Table 6: Additional ablation on DTU dataset, reporting PSNR for each scene. (a): the fixed densification (ADC) algorithm, (b): erank regularization.

| Method | 24 | 37 | 40 | 55 | 63 | 65 | 69 | 83 |
|---|---|---|---|---|---|---|---|---|
| 3DGS | 30.45 | 26.93 | 29.79 | 31.92 | 35.42 | 31.09 | 28.34 | 38.00 |
| +a | 30.69 | 27.14 | 30.31 | 32.01 | 35.93 | 31.23 | 28.04 | 37.95 |
| +a+b | 30.90 | 27.21 | 30.42 | 32.23 | 35.81 | 31.62 | 28.41 | 38.00 |

| Method | 97 | 105 | 106 | 110 | 114 | 118 | 122 | Mean |
|---|---|---|---|---|---|---|---|---|
| 3DGS | 30.20 | 34.32 | 35.00 | 34.65 | 30.86 | 37.25 | 38.07 | 32.82 |
| +a | 30.25 | 34.30 | 35.11 | 34.59 | 31.10 | 37.65 | 38.21 | 32.97 |
| +a+b | 30.27 | 34.41 | 35.22 | 34.69 | 31.20 | 37.69 | 38.23 | 33.09 |

Splatting [35]. Combined with the inherent implicit shrinkage bias of 3DGS [16, 35], this results in the underestimation of the scale parameters during the optimization process.

Second, the densification along the longer axis does not occur effectively since the longer axes, or the axes with large variance, have smaller gradients. When Gaussians move in the direction of the shorter axis, pixel values change abruptly. In contrast, there are only small changes in pixel values when moving along the longer axis. Specifically, when $\frac{\partial \mathbf{p}_i}{\partial \mathbf{u}}$ aligns with the direction of the longest axis, the gradient values are typically small. Consequently, the norm of the final gradient often falls below the densification threshold $\|\frac{\partial L}{\partial \mathbf{x}}\|_2 < \tau_\mathbf{x}$, preventing effective densification. We visualize $\frac{\partial \mathcal{G}_k}{\partial \mathbf{x}}$ in arrows in Fig. 8 (a), which is proportional to $\frac{\partial \mathbf{p}_i}{\partial \mathbf{u}}$, for better understanding. Therefore, the splats are biased towards adjusting their scale parameters (Fig. 8 (b)) rather than splitting along the longer axis, converging into needle-like Gaussians.

Third, scale parameters are kept the same after splitting, so needles are not shortened after densification.

It will be interesting future work to delve deeper into these reasons and address the problem with other approaches.

### A.7 Additional qualitative results

We present a normal rendering of our method results. Fig. 9 are results of the scene 122, with the depth distortion and normal regularization loss used together. Fig. 10 shows the results of scene 55. Fig. 11 shows the rendering results of the Mip-NeRF360 dataset of our method. We visualize the Gaussians with an effective rank smaller than 1.02 in red. The effective rank regularization is adaptive to the scene, reducing the number of needle-like Gaussians, while effectively representing the required regions.

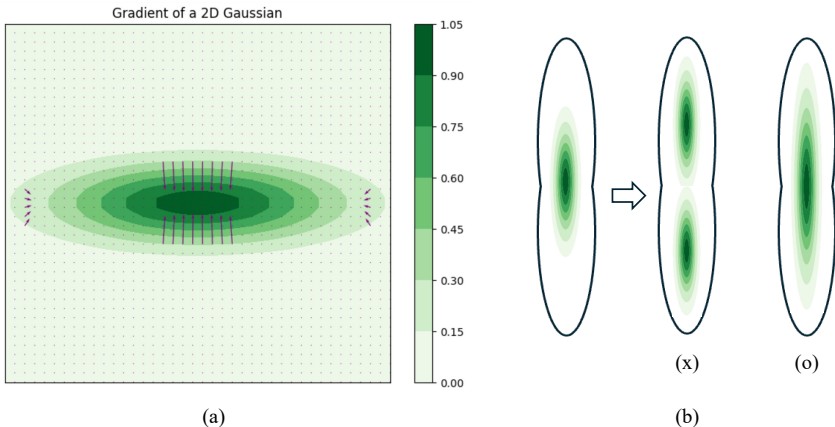

Figure 8: (a): Visualization of $\frac{\partial \mathcal{G}_k}{\partial \mathbf{x}}$ in arrows, which is proportional to $\frac{\partial \mathbf{p}_i}{\partial \mathbf{u}}$. (b) The splats are biased towards adjusting its scale parameters rather than splitting along the longer axis, converging into needle-like Gaussians.

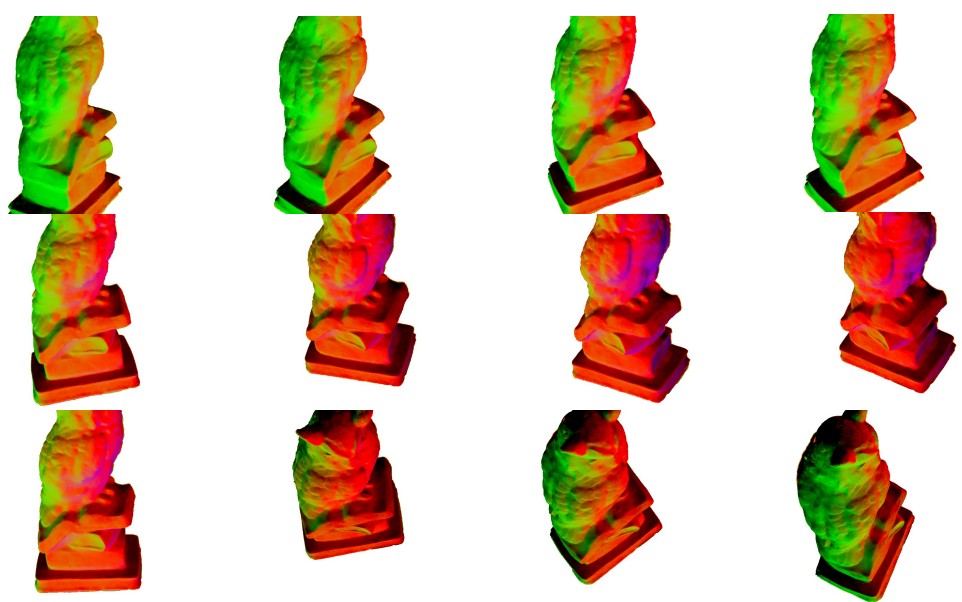

Figure 9: Normal rendering results of DTU dataset (scene 122) of our method, with depth distortion and normal regularization loss.

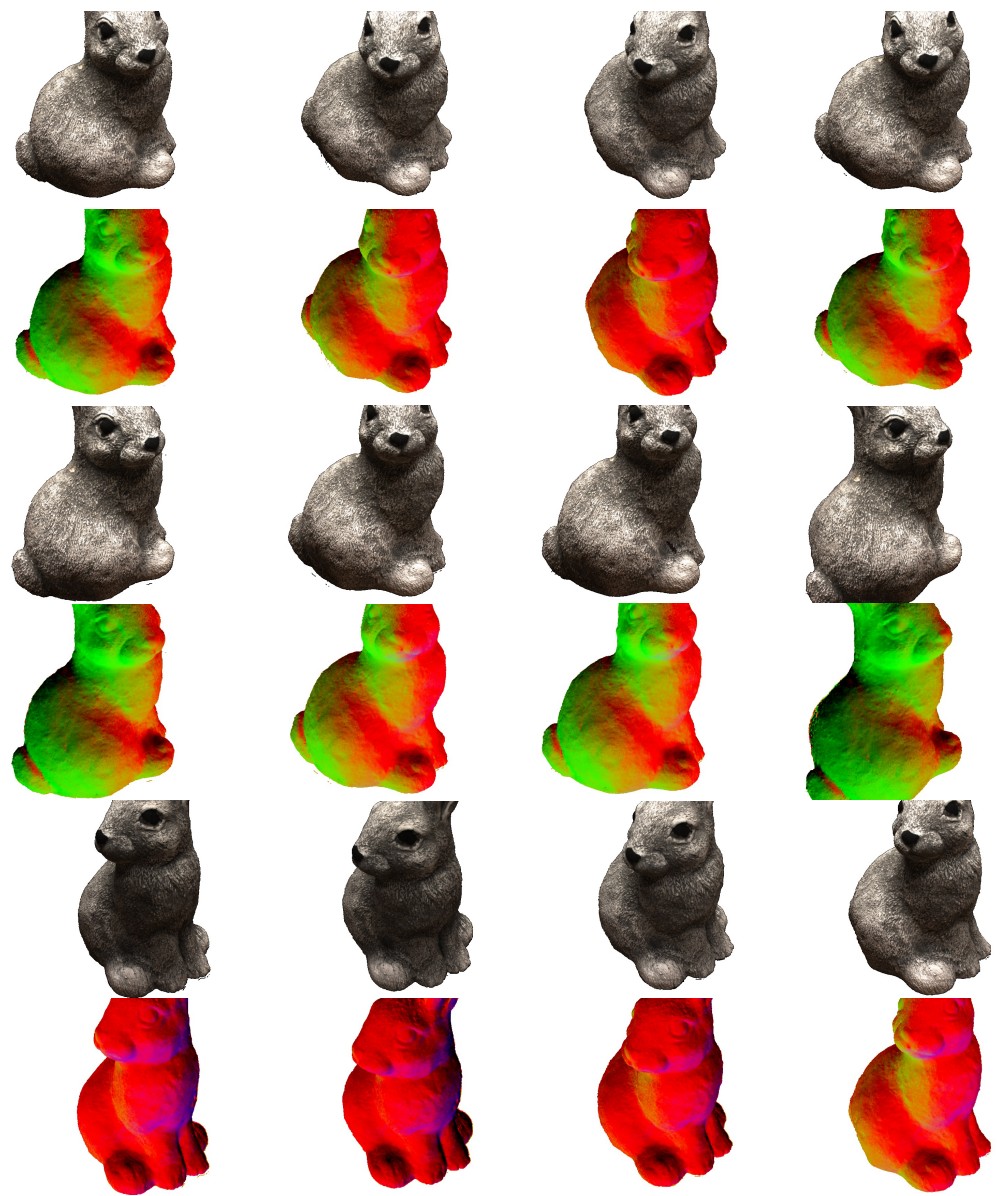

Figure 10: Normal rendering and visual rendering results of DTU dataset (scene 55) of our method, with depth distortion and normal regularization loss.

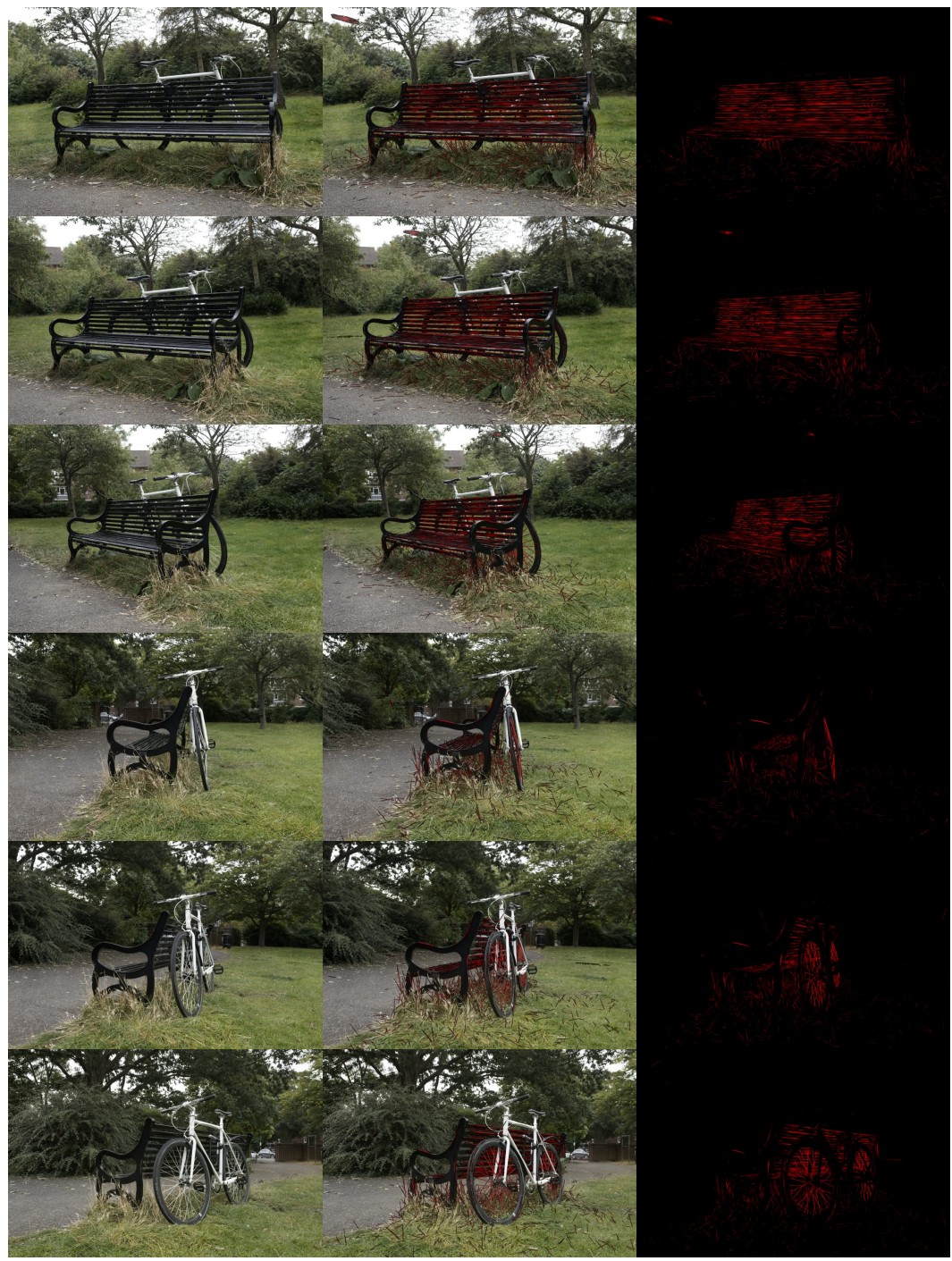

Figure 11: Rendering results of Mip-NeRF360 dataset of our method. We visualize Gaussians with an effective rank smaller than 1.02 in red. The effective rank regularization is adaptive to the scene, reducing the number of needle-like Gaussians, while effectively representing the required regions.

