# OpenReview forum: "Effective Rank Analysis and Regularization for Enhanced 3D Gaussian Splatting"
_NeurIPS.cc/2024/Conference — NeurIPS 2024 poster_

### Official Review · Reviewer_n78p · 2024-06-27

**Soundness:** 3
**Presentation:** 3
**Contribution:** 2
**Rating:** 5
**Confidence:** 5

**Summary:**

The authors present a new regularization for Gaussian splatting (GS) method that increases the Shannon entropy of the scale parameters (it was named erank as while rendering the scale is presented as a diagonal matrix, and the aforementioned calculation turns into the effective rank), to better reconstruct 3d mesh out of the GS point cloud. The paper argues that a majority of the gaussians in previous methods are close to erank of 1, which ruins the 3d reconstruction because of needle-like gaussians that effectively add noise to the reconstructions. Given that regularization as an add-ons to other methods it improves the 3D reconstruction, as shown in the experiments on DTU dataset. Additionally, it shows in the appendix that the image rendering abilities of the GS is not impaired because of the added regularization.

**Strengths:**

The paper dives into the attributes of the GS and impressively recognizes an issue relating to the GS ability to reconstruct 3d mesh. The authors exhibit an analysis of the amount of 1-ranked gaussians that strengthen their claim, and showed improved results on DTU dataset.

**Weaknesses:**

There are several concerns regarding this paper, first it should include more benchmarks for 3d reconstruction, for example Tanks and Temples dataset which was used as benchmark in many other works in the field, including the works that was cited in the paper. Secondly, there are two works that the authors did not mention [1,2] and should be compared to in the DTU experiment. The first [1] has the same idea as 2DGS and was already published. It might be that their regularization solves part of the issue of the GS that this paper presents. The second [2], does not use the gaussian splatting directly for reconstruction and instead renders stereo images for depth estimate. Thus, it does not share the reconstruction error because of the needle-like gaussians. Comparing both of them will strengthen this paper. Lastly, there is no adequate justification for the erank regularization in comparison to other possibilities, e.g. linear, parabolic, or exponential loss on the size of the smallest magnitude scale. Finally, an ablation study showing why erank-regularization is superior compared to other losses is missing.
[1] Dai, P., et al. "High-quality surface reconstruction using gaussian surfels." arXiv preprint arXiv:2404.17774 (2024).
[2] Wolf, Y., et al.. "Surface Reconstruction from Gaussian Splatting via Novel Stereo Views." arXiv preprint arXiv:2404.01810 (2024).

**Questions:**

see weaknesses

**Limitations:**

see weaknesses

---

> ### Author Rebuttal · Authors · 2024-08-07
>
> We appreciate the reviewer for acknowledging the contribution of our work. We are grateful for the helpful reviews that could strengthen our paper.
> ### Tanks and Temples
> >Thank you for the suggestion. We conducted the experiments after the submission, and the table below shows a general improvement when our method is used as an add-on module. Also we want to note that, unlike other papers, we needed four times more experiments because we tested our method as an add-on module on four different models: 3DGS, SuGaR, 2DGS, and GOF. For a single DTU evaluation alone, we ran 15 scenes * 5 repeated experiments per scene * 4 models, totaling 300 training and evaluation runs, which took an extremely long time.
>
> | Methods | 3DGS | 3DGS+e | GOF | GOF+e |
> |:--------:|:--------:|:--------:|:--------:|:--------:|
> | Barn |   0.13 |   0.32 | 0.51 | 0.61 |
> | Caterpillar |  0.08 |  0.19 | 0.41 | 0.42 |
> | Courthouse |  0.09 |  0.13 | 0.28 | 0.28 |
> | Ignatius |  0.04 |  0.37 | 0.68 | 0.74 |
> | Meetingroom |  0.01 |  0.15 | 0.28 | 0.30 |
> | Truck |  0.19 |  0.24 | 0.59 | 0.59 |
>
>
> ### Related Works
> >Thank you for suggesting relevant papers. We appreciate the recommendations to strengthen our paper. We were aware of these papers and are also curious about the results. However, we believe these works are concurrent, and not comparing them should not be considered a weakness. Notably, [2] did not have code released at the time of our submission (the code was recently released, during the rebuttal period), and it was not published when we submitted to NeurIPS (ECCV24's final decision was in July). Therefore, it was not possible to compare it with our work. Similarly, [1] was published on arXiv on April 30 (initial version on April 27), and our final draft for NeurIPS was completed around the first week of May. Still, we believe we have shown our efforts to track and include the latest papers, GOF was on arXiv on April 16, and 2DGS on March 26. We are willing to conduct more experiments and share results during the discussion period (if requested, following the rules of discussion period), and the rest in the camera-ready version of our paper.
> ### Justification of Using Effective Rank Regularization
> >- Interpretability: Effective rank provides interpretable and meaningful numbers for regularization. Using other variants (linear, exponential) would yield different values without explicit meaning. We know that Gaussians with erank(G)=2 is disk-like, erank(G)$\approx$1 is needle-like, and we aim to eliminate non-disk-like Gaussians (erank(G) > 2) or needle-like Gaussians (erank(G) $\approx$1).
> > - Comprehensive Regularization: Our regularization considers all three axes. Some works, including PhysGaussian, have tried regularizing Gaussian primitives but do not focus on all three axes. PhysGaussian considers two axes (max and min scale axes) to reduce spiky Gaussians, but not enforcing disk-like Gaussians. The attached PDF Fig.3(b) shows the erank histogram of PhysGaussian, where it reduces spiky Gaussians (erank(G) $\approx$ 1) but does not enforce disk-like Gaussians (erank(G) = 2) or penalize Gaussians with erank(G) > 2. Methods like GaussianShader only consider the axis with the minimum scale, minimizing it to make the Gaussian primitive “flat”. 2DGS uses 2D surfel as a primitive, achieving similar effects but not handling spiky Gaussians. Our method considers all three axes, enforcing disk-like Gaussians without needle-like ones, using effective rank.
> >- Elegant Loss Definition: Effective rank of Gaussian covariance enables a single scalar term representation of the Gaussian geometry, which is an elegant way to define loss, avoiding the need to consider all three combinations (s1-s2,s2-s3,s1-s3) of three axes. Additionally, logarithmic loss is known for its stability in optimization problems (we emprically prove this below).
>
> ### Comparison & Ablation Study
> >We conducted an ablation study on different variants as per your suggestion. We denote s1, s2, s3 as the largest, second largest, and smallest scale.
> >- vs. PhysGaussian (minimize s1 / s3)
> >- vs. linear (minimize s1 / s2, minimize s3)
>
> | DTU scan | 24 | 37 | 40 | 55 | 63 | 65 | 69
> |:--------:|:--------:|:--------:|:--------:|:--------:|:--------:|:--------:|:--------:|
> | 3DGS |   2.14 |   1.53 | 2.08 | 1.68 | 3.49|2.21|1.43|
> | 3DGS+e |   0.85 |  0.77 | 0.88 | 0.51 | 1.21|1.45|0.96|
> | PhysGaussian |   0.87 |   0.81 | 0.86 | 1.36 | 2.99|1.97|1.46|
> | 3DGS+linear | 0.89   | 0.80   |0.91  | 1.21 | 2.03| 1.84| 1.44|
>
>
> >The results (DTU scan, Chamfer distance $\downarrow$) show that our method is superior to other variants, empirically proving our justification.
>
>
>
> Once again, we appreciate the reviewer for taking the time to read through the paper and our response. Your reviews are immensely helpful in strengthening our work. We will add more results (including the supplemental video) during the discussion period.
> Thank you.

---

> > ### Comment · Reviewer_n78p · 2024-08-11
> > **Comparison with recent results would improve the paper.**
> >
> > The authors provided answers to our concerns regarding the T&T dataset, and the justification for using the entropy like regularization compared to other possibilities. Regarding the recent papers to compare with, such evaluations would definitely improve the paper.

---

### Official Review · Reviewer_7LDG · 2024-06-30

**Soundness:** 3
**Presentation:** 4
**Contribution:** 3
**Rating:** 7
**Confidence:** 4

**Summary:**

3D Gaussian Splatting is a remarkable technique in novel view synthesis. However, it usually degenerates into noodle-like shapes which sometimes bring visual artifacts and inaccurate geometry. This paper analyzes the phenomenon and proposes an effective rank loss to regularize the Gaussians. The technique seems to be a plug-and-play module for 3DGS variants. Extensive experiments suggest the effectiveness.

**Strengths:**

1. The observation and analysis are strong and the solution of using effective rank is simple yet effective.
2. The results show clear improvements over representative works including 3DGS, SuGaR, and 2DGS, which cover a broad range of scene representations.
3. The paper is well-written and easy to follow. I have no issue with the written. Most of the related works are adequately discussed.
4. The technical details are sufficient and I believe an expert in 3DGS can easily implement it.

**Weaknesses:**

1. There is no supplemental video, which is essential for nvs tasks.
2. The improvement of e-rank for Novel view synthesis is limited. It would be beneficial to highlight how this could improve novel view synthesis.
3. Although the paper analyzes that the noodle-like artifacts are due to the training bias, they can also be created to represent high-frequency details. The paper should highlight the PSNR in Figure.1 to give readers a sense that such regularization indeed improves novel view synthesis.
4. It would be beneficial to add the number of Gaussians since enforcing the Gaussian to 2D from 1D should reduce the number of Gaussians in representing a scene.

**Questions:**

1. How the normal is constructed for 3D Gaussian and how it is visualized?
2. What is the effect of densification on geometry reconstruction in 3DGS? In Table 1. Does 3DGS+e indicate with ERANK only or plus densification? It should be clearly ablated where the performance comes from.

**Limitations:**

Overall, the paper did a good job with minor issues required to clarify. The limitation of tuning the hyperparameter is also discussed and the potential negative effect is reasonable.

**minors:**

The normal visualization seems not to be normalized witn [0,1]

---

> ### Author Rebuttal · Authors · 2024-08-07
>
> We appreciate the reviewer for acknowledging the effectiveness and simplicity of our method. Additionally, we are grateful for pointing out the strong analysis and concise writing.
>
> ### Supplemental Video
> >We promise to share the video results in the project page. Please stay tuned!
>
> ### Novel view synthesis
> >The primary focus of our paper is geometry reconstruction, which we evaluate using Chamfer Distance, rather than novel view synthesis. Unlike other geometry reconstruction works that has geometry and visual quality tradeoff, our method maintains high visual quality without compromise, which we later found is an additional strength of our model. Also it reduces spiky artifacts in novel views. In constrained scenes with well-prepared data, these spiky artifacts appear in very small regions, potentially explaining the limited improvement in visual quality metrics. However, in few-shot settings and renderings from extreme viewpoints (far from the training view), we observe larger needle artifacts. It would be interesting to see results with such datasets.
>
> ### Figure 1 Visibility
> >Thank you for pointing out the visibility issue in Figure 1. We will improve the figure as per your suggestion. You are correct that anisotropic Gaussians are necessary for representing high-frequency details. The method should, therefore, only encourage a tendency towards disk-like Gaussians as regularization, rather than “eliminating” them.
>
> ### Adding the number of Gaussians
> >We appreciate your idea. Actually, we had similar thoughts and have tested several related settings previously. Specifically, we tried lowering the densification threshold (causing more frequent densification) and lowering the pruning threshold (resulting in less frequent pruning). Neither of these experiments improved the metrics. Surprisingly, they resulted in worse PSNR in test views, despite higher PSNR for training views. This indicates that more Gaussians may lead to overfitting of the training views without improving the visual quality of novel test views. Our method, in fact, reduces overfitting, as evidenced by generally lower training PSNR but higher PSNR for testing views, as presented in the main paper.
>
> ### Q: Normal Calculation
> >There are several ways to calculate the normal. “Depth normal” is derived from 2D rendered depth and is used for depth normal consistency loss (used in 2DGS and GOF). For individual Gaussians, 3DGS and SuGaR use the shortest axis as the normal. GOF uses ray-dependent normal (the normal of the intersection plane; please refer to the GOF paper for more details). These normals are rendered into screen space using the same alpha blending method used to render the RGB image.
>
> ### Q: Ablation Studies
> >Please refer to Table 2 of the main paper for ablation results. Both erank regularization and densification bring performance gains. (+e indicate both).
>
> We again appreciate the reviewer for taking the time to read through the paper and this lengthy response. Your reviews are immensely helpful in strengthening our work. We will add more results (including the supplemental video) during the discussion period.
> Thank you.

---

> > ### Comment · Reviewer_7LDG · 2024-08-09
> > **Official Comment by Reviewer 7LDG**
> >
> > Thank you for your response. I appreciate that most of my questions have been addressed. After reading the author rebuttal and considering the other reviews, I’d like to share my thoughts.
> >
> > The proposed method is simple yet effective and can serve as a plug-and-play module for various GS variants, potentially broadening its impact.
> >
> > I also agree with Reviewer w1Sn that there should be more analysis on the root cause of the needle-like artifacts, as this could enhance the paper’s impact. I encourage the authors to include some of this analysis in the main paper. Additionally, it would be beneficial to emphasize that the regularization, while reducing needle-like artifacts, does not compromise the reconstruction of high-frequency details.
> >
> > I also noticed, as pointed out by Reviewer biCn, that some important baselines like PhysGaussian were missing in my initial feedback. I appreciate that the authors have since provided these comparisons.
> >
> > Overall, while some minor revisions are needed, I believe the authors are well-equipped to address these issues, as demonstrated in their rebuttal, including the comments and tables presented. Therefore, I have no concerns with the majority of the work and would like to maintain my original rating.

---

### Official Review · Reviewer_w1Sn · 2024-07-09

**Soundness:** 3
**Presentation:** 3
**Contribution:** 1
**Rating:** 5
**Confidence:** 4

**Summary:**

This paper performs a statistical analysis on 3DGS for its effective rank distribution of the learned Gaussians. It claims that most of the Gaussian learned are close to rank 1 effectively, giving needle-like artifacts in novel view synthesis and reconstruction. Hence, this paper proposes a regularization loss to discourage low range Gaussians from forming, showing improved reconstruction results.

**Strengths:**

**Motivation**
* This paper starts with a clear and strong motivation, where the Gaussian learning with 3DGS creates needle-like artifacts. This reduces both the reconstruction and novel view synthesis quality.

**Method**
* This paper proposes an intuitive and effective method of regularizing the effective rank of the Gaussians, to regularize on the shape of individual Gaussian. The experiment results also demonstrate the effectiveness of the proposed method on various baselines, with considerable improvement in reconstruction results and minor improvement in novel view synthesis results.

**Weaknesses:**

I do not doubt the motivation and effectiveness of the proposed method. In fact, needle-like artifacts have been a well-known issue for 3DGS in the community. However, I feel reluctant to accept this paper due to the following reasons:
1. Although the phenomenon is well known by many, the root cause of this issue is not clear. This paper tries to explain the cause in its appendix, and gives two hypotheses: a) incorrect dilation observed by MipGaussian, b) Gaussians are more likely to densify along their short axis instead of their long axis. Based on this, I would like to point out the following issues:
- This part should be put into the main paper instead of the statistical analysis. The causes of the low-rank Gaussians would be much more valuable than reiterating the phenomenon that many people already know.
- This cause analysis needs to be more rigorous. The version provided in the appendix is more of a hypothesis than an analysis.
- If it is because of the dilation error, is it already fixed by MipGaussian? What is the rank distribution like for MipGaussian? Can the proposed method improve the MipGaussian performance as well? If yes, then something else is still causing the issue. If not, why shouldn't MipGaussian be used instead of the proposed method?
- If it is because of the densification bias, does this problem still occur without densification at all? In fact, I have personally observed many cases of needle-like artifacts that are so long that extend across the entire image. In this case, the rendering loss should be enough to prevent this from happening without anything to do with the densification. But it is not happening, why? Is there something wrong with the rendering loss supervision we have with 3DGS? I am not saying that the densification bias is not a cause, but it requires more evidence and experiment results to be proved.

2. Continuing from the previous point, the method proposed does not address the cause hypothesized. It is not fair to say that regularization methods are just bad, but there have been even simpler regularization methods used by other papers (e.g. PhyGaussian). The most common method is to regularize the ratio between the long axis and the short axis of the 3DGS. It takes only a few lines of code and works reasonably well. This paper does not mention this simpler regularization and does not compare with it as a baseline. However, I think it is very important to justify the more complicated method proposed in this paper, with either rigorous analysis as well as empirical results (preferably both).

In summary, I believe that this paper lacks a rigorous analysis of the cause of the low-rank Gaussian formation, and the proposed method lacks novelty and performance comparison against the simple naive regularization used in other papers.

**Questions:**

To summarize my opinions mentioned in the weakness section into questions, it would be:
1. what is really the cause for the low-rank Gaussian formation and how does the proposed method address it?
2. why is the proposed method better than the simple regularization on the ratio of the long/short gaussian axis?

**Limitations:**

The paper does not have potential negative societal impact and there is no significant limitation of the method that needs to be addressed other than the ones mentioned in the previous sections.

---

> ### Author Rebuttal · Authors · 2024-08-07
>
> We appreciate your high-quality review and detailed understanding of our work. We are also grateful for acknowledging the motivation and effectiveness of our method.
>
> We agree that shedding light on and analyzing the causes of low-rank Gaussians would add value to our paper. Therefore, we conducted additional experiments, including 2D toy experiments, Mip-Splatting (MipGaussian) analysis, and comparisons with PhysGaussian. Please refer to the general comment for the 2D toy experiment Mip-Splatting and reviewers #1 and #4 for the comparison with PhysGaussian and its variants.
>
> While we conducted all the experiments, we also would like to explain our approach and the reasoning behind it. Specifically, we want to address the points about the lack of “rigorous analysis of the cause of the low-rank Gaussian” and the “method not addressing the cause.” While a detailed analysis of the root cause is one way to approach the problem, we believe our approach of writing a paper is also valid.
>
> For instance, consider a scenario where low test accuracy (phenomenon) is caused by overfitting due to insufficient data (root cause). One could address this by using more training data. However, directly addressing the phenomenon and focusing on improving test accuracy through different techniques and regularizations (without increasing data) is also a plausible approach. Many impactful papers focus on solving phenomena without rigorous root cause analysis, and they significantly contribute to the field. For example, Mip-NeRF 360, focuses on solving phenomena (floater artifact) without rigorous root cause analysis.
> Our approach is similar. We had a clear target for the optimal shape of Gaussian primitives—disk-like (flat) and non-needle-like. This made our direct approach to tackling the phenomenon more straightforward and convincing.
>
> We do not argue that our approach is the best, but we believe it is a plausible one. Therefore, we believe that the focus should be on evaluating the logical and technical aspects of our paper rather than suggesting an alternative approach.
>
> Still, we tried our best to analyze the cause of needle-like Gaussians, presented in general rebuttal section above.
>
> Additionally, we do not believe we are reiterating known phenomena. While many have observed spiky (needle-like) Gaussians and some papers, like PhysGaussian, address them, their impact on geometric reconstruction (evaluated with Chamfer Distance) has not been thoroughly explored. We are the first to introduce the tool of effective rank to 3DGS, enabling `interpretable` statistical evaluation of the shapes of individual Gaussians. We discovered that spiky or non-disk-like shapes of Gaussians significantly impact the geometric quality of the reconstructed scene, beyond just causing artifacts. This focus on geometric quality and analysis is a new contribution.
>
> Moreover, no previous works have focused on all three axes of the Gaussians. For more details on this aspect, please refer to our responses to reviewers #1 and #4.
>
>
> ## The cause of spiky Gaussians
> > Please refer to the general comment
>
> ## Comparison and justification with other variants & novelty
> > Please refer to our responses to reviewers #1 (`PhysGaussian` and `vs. PhysGaussian and variants` section) and #4 (`Justification of Using Effective Rank Regularization` and `Comparison & Ablation study` section).
>
>
> Thank you once again for your valuable feedback.

---

> > ### Comment · Reviewer_w1Sn · 2024-08-08
> >
> > I appreciate the effort put in by the authors to conduct the toy experiment and the comparison against scaling loss from PhyGaussian. I am convinced that the low dL/du gradient problem demonstrated in the toy example is partly the reason why the Gaussians tend not to split along the long axis. Hence, I would like to raise my score. I hope the additional experiments and justifications can be added to the final version.
> >
> > However, I encourage the authors to look deeper into the scaling optimization of Gaussians as well. Other than the reason shown, there should be something about the scaling optimization of Gaussians also causing the spiky Gaussians, because we often see very long and thin Gaussians that should have been penalized heavily on rendering loss even from training views. This phenomenon can hardly be explained by splitting bias alone. I believe that finding out the reason behind this can be very benefitial to the research community.

---

### Official Review · Reviewer_biCn · 2024-07-10

**Soundness:** 3
**Presentation:** 3
**Contribution:** 2
**Rating:** 6
**Confidence:** 4

**Summary:**

The paper identifies a common problem in 3D Gaussian splatting where Gaussians converge to needle-like shapes, with its variance mostly contained in one axis. These needle-shaped Gaussians can cause artifacts and inaccurate surface reconstruction.
To quantify this phenomenon, the paper uses the concept of effective rank, a continuous generalization of the rank of a matrix.  During scene reconstruction, while 3D gaussians may begin with an effective rank of 3, they tend to converge to an effective rank of 1 after several thousand iterations of training.  Based on effective rank, the paper proposes a regularization method to reduce needle-shaped Gaussians. On the DTU dataset, the authors show that 3DGS with rank regularization causes better chamfer distance and PSNR metrics.

**Strengths:**

The paper introduces a method to prevent needle-like Gaussians in 3DGS, an important challenge in view synthesis, and presents a clear way to quantify the problem. The authors propose a new regularization term that is simple and can be used alongside other Gaussian splat variants such as 2DGS and SuGaR. On the DTU dataset, the authors report better chamfer distance and PSNR metrics than prior work.

Overall I found the paper very interesting and its claims well justified. Because of the widespread popularity of Gaussian Splatting, the method can be useful to both researchers and 3D designers.

**Weaknesses:**

While the paper introduces a new way to quantify needle-like Gaussians, this is not the first paper to propose a regularization term for them.  PhysGaussian [A] introduces an anisotropic regularizer to prevent skinny Gaussians. This regularization term has already been implemented in Nerfstudio’s Splatfacto method [B]. This seems like the most relevant and important comparison to the proposed method, and is completely missing from the paper. This missing comparison is the main reasoning behind my score.

In addition, I wish that the paper did a deeper investigation into what causes spikey Gaussians. I found the discussion in Appendix A.4. very fascinating, and think it could be very valuable to the community if elaborated on. Maybe a simple experiment validating the hypothesis could strengthen the paper.

Minor Comments:
I found that Fig. 1 was hard to parse. I think it would help to include labels for the figure’s images.

[A] https://arxiv.org/pdf/2311.12198
[B] https://docs.nerf.studio/nerfology/methods/splat.html

**Questions:**

What happens if you do not regularize erank(G) < 2? Do Gaussians with erank(G) > 2 hurt reconstruction quality?

What are x and o in Fig. 8?

**Limitations:**

Yes the authors have discussed the limitations and potential societal impacts.

---

> ### Author Rebuttal · Authors · 2024-08-07
>
> We appreciate the reviewer for highlighting the importance of the task, as well as the effectiveness and clarity of our method. We are also grateful for suggesting potential baselines and ablations to strengthen our work.
>
> ### PhysGaussian
> >Initially, we did not consider PhysGaussian due to its focus on physics-grounded deformation, with regularization primarily aimed at reducing spiky artifacts caused by deformation. However, we acknowledge the similarity between our method and PhysGaussian and agree that it could be used to enhance 3D geometry. We have conducted additional experiments using PhysGaussian’s regularization and plan to include these results in the camera-ready version of our paper (possibly in the appendix).
>
> ### vs. PhysGaussian and variants
> >Before presenting the results, we want to clarify that PhysGaussian’s regularization method differs from ours. Some works, including PhysGaussian, have tried regularizing Gaussian primitives, but none focus on all three axes of the Gaussians. PhysGaussian considers two axes (max and min scale axes) to remove spiky Gaussians, which reduces spiky Gaussians but does not enforce disk-like (flat) Gaussians. We present the erank histogram of PhysGaussian in the attached PDF Fig. 3 (b), showing reduced spiky Gaussians (erank(G) $\approx$ 1) but not explicitly enforcing disk-like Gaussians (erank(G) = 2) or penalizing Gaussians with erank(G) > 2.
> Additionally, methods like GaussianShader only consider the axis with the minimum scale, minimizing it to make the Gaussian primitive “flat”. 2DGS uses 2D surfel as a primitive, achieving similar effects. But these methods does not handle spiky Gaussians. Our method accounts for all three axes, enforcing disk-like Gaussians without needle-like Gaussians, using effective rank. This approach leads to better quantitative results, as shown in the table below.
>
> > Also, our approach provides interpretability of the Gaussian shapes via effective rank (erank(G)=3: sphere, erank(G)=2: disk, erank(G)=1 needle), and logarithmic loss is known for its stability in optimization problems (we emprically prove this in below table, also refer to results in reviewer #4). Please refer to the response to reviewer #4 for more justification and experimental results.
>
> ### deeper investigation into what causes spiky Gaussians
> > please refer to general rebuttal response above.
>
>
> ### Additional comments
> >- We appreciate the comment and will update Fig. 1 with labels.
> >- Yes, non-disk-like Gaussians (erank(G) > 2) negatively impact the geometry metric. Prior to submission, we conducted various experiments targeting erank(G) = x, where x > 2, and consistently found that the results were worse compared to our method. As an example, we present a case where the loss is applied with a target erank(G) = 2.5. The results are shown in the table below.
> >- In Fig. 8(b) of the main paper, we are showing a toy experiment where a Gaussian is not split into two Gaussians (x), and instead, the scale is adjusted (o), which is not optimal in this case. We apologize for the confusion and will improve the figure and caption. We provide more 2D toy experiment in the attached PDF.
>
>
> | DTU scan | 24 | 37 | 40 | 55 | 63 | 65 | 69
> |:--------:|:--------:|:--------:|:--------:|:--------:|:--------:|:--------:|:--------:|
> | 3DGS |   2.14 |   1.53 | 2.08 | 1.68 | 3.49|2.21|1.43|
> | 3DGS+e |   0.85 |  0.77 | 0.88 | 0.51 | 1.21|1.45|0.96|
> | PhysGaussian |   0.87 |   0.81 | 0.86 | 1.36 | 2.99|1.97|1.46|
> | 3DGS+erank(G)=2.5 |  1.08  | 1.26   | 1.34 | 0.97 |2.34 |2.31|1.06|
>
> Thank you again for your valuable feedback.

---

> > ### Comment · Reviewer_biCn · 2024-08-09
> > **Thanks**
> >
> > Thank you for your response. I appreciate the new experiments on PhysGaussian, and I see how their method can still cause low-rank Gaussians from forming. Along with the other reviewers, I agree that a deeper investigation into the cause of spiky Gaussians is important for the paper, and I hope that the new analysis is included in a revised version of the paper.
> >
> > Based on the rebuttal, I will change my score.

---

### Author Rebuttal · Authors · 2024-08-07

We thank the reviewers for acknowledging the effectiveness of our work and highlighting the importance of the task. We are also grateful to all the reviewers for taking the time to read through the paper and providing detailed feedback. Your reviews are immensely helpful in strengthening our work.

We have responded to each review individually and use this general rebuttal space to share more details about the cause of needle-like Gaussians, as requested by reviewers #1 and #2.

While investigating and analyzing the cause of spiky Gaussians is not the main contribution of our work (thus it is included in the appendix), we do agree that further analysis on this aspect is interesting and could potentially strengthen our work. More discussion on this is provided in our response to review #2.

We suggested three reasons for the causes of needle-like Gaussians in the paper: 1. Dilation, 2. Densification trigger along the longer axis, and 3. unadjusted scale after densification.

### (1) Mip-Splatting and Dilation Operation

> As requested by reviewer #2, we first present the Mip-Splatting (MipGaussian) erank histogram in the attached PDF, Fig.3 (a), along with the Chamfer Distance metric (table below). To clarify, we did not claim that the dilation operation solely causes needle-like Gaussians. The dilation operation, combined with the implicit shrinkage bias (cite) of 3DGS, can cause the scale of some axes to be small. Densification trigger issue (2) causes needle-like Gaussians, and dilation can further boost this phenomenon by making the smallest scale even smaller.

>Mip-Splatting focuses only on Gaussians smaller than the pixel size, which has a negligible impact on effective rank measurement and geometry reconstruction. For instance, given two needle-like Gaussians with 10x scale differences: scales (1,0.01,0.01) and (1, 0.001, 0.001), the eranks are 1.002 and 1.000, respectively, indicating a small difference.

>Moreover, Mip-Splatting does not constrain Gaussians to be flat (disk-like), which is crucial for geometry reconstruction. The results show that Mip-Splatting produces a similar effective rank distribution as the original 3DGS and does not improve geometry reconstruction. Thus, while dilation may contribute to the formation of needle-like Gaussians, it is not the main reason, and its impact is small.

| DTU scan | 24 | 37 | 40 | 55 | 63 | 65 | 69
|:--------:|:--------:|:--------:|:--------:|:--------:|:--------:|:--------:|:--------:|
| 3DGS |   2.14 |   1.53 | 2.08 | 1.68 | 3.49|2.21|1.43|
| 3DGS+e |   0.85 |  0.77 | 0.88 | 0.51 | 1.21|1.45|0.96|
| Mip-Splatting |   2.45 | 2.21 | 1.66 | 1.51 | 3.28 |2.32| 1.46|


### (2) Densification Trigger
>Densification issue 2 is straightforward. The densification trigger is the norm of $\sum_{i \in \mathcal{P}} \frac{\partial L}{\partial p_i} \frac{\partial p_i}{\partial u}$ larger than the densification threshold, and $\frac{\partial p_i}{\partial u}$ is small when u moves in the direction of the longest axis (Gaussians have small gradients along the longer axis). Thus, densification along the longer axis is difficult to trigger.


### (3) Scale After Densification
> When a Gaussian is split along the longest axis, instead of halving the largest scale, it is kept the same (scale is copied), resulting in two spiky Gaussians.

### Experiments
>We conducted toy experiments to empirically demonstrate that:

>Densification along the longer axis is less preferred, and Gaussians tend to elongate, even when initialized with multiple Gaussians.

>Fig. 1 and 2 in the PDF show a 2D toy setting of Gaussian splatting. The first row indicates the target image (left) and the initial Gaussian(s) (right). The second row shows the fitted Gaussian(s) and the absolute difference between the target and the fitted.

>Fig. 1 (a) in PDF suggests that Gaussians are not properly densified when they should densify along the longer axis. We also numerically observe that dL/du is very small in this case, hindering densification along the longer axis compared to (b). Lowering the densification threshold might be suggested, but this increases densification in all directions, which does not reduce the spikes. Also in practice, it is not feasible because the number of Gaussians would explode even with a small decrease in the threshold. For example, as presented in Fig. 3 (c) of the PDF, lowering the threshold from 0.0002 to 0.0001 results in twice as many Gaussians, increasing training or GPU memory requirements and training time.

>Similarly, Figure 2 (a) and (b) of the PDF suggest that Gaussians tend to elongate instead of splitting, even when initialized with multiple Gaussians.

>Our method addresses issue #2 by limiting the anisotropic Gaussians, but tackling issue #3 and modifying the scale after splitting would be an interesting future work.

---

### Decision · Program_Chairs · 2024-09-25

**Decision:**

Accept (poster)

**Comment:**

This paper was reviewed by four experts in the field.  Based on the reviewers' feedback, the decision is to recommend the paper for acceptance.  The reviewers did raise some valuable concerns that should be addressed in the final camera-ready version of the paper. The authors are encouraged to make the necessary changes to the best of their ability.    We congratulate the authors on the acceptance of their paper!